# Effects of Suryanamaskar, an Intensive Yoga Exercise Routine, on the Stress Levels and Emotional Intelligence of Indian Students

**DOI:** 10.3390/ijerph20042845

**Published:** 2023-02-06

**Authors:** Krzysztof Stec, Marek Kruszewski, Leon Ciechanowski

**Affiliations:** 1Faculty of Health Sciences, University of Czestochowa, 42-200 Czestochowa, Poland; 2Physical Education Department, Jozef Piłsudski University of Physical Education, 00-968 Warszawa, Poland; 3Department of Management, Kozminski University, 03-301 Warsaw, Poland; 4Department of Psychology, University of Social Sciences and Humanities, 03-815 Warsaw, Poland

**Keywords:** yoga, Suryanamaskar, mental health, emotional intelligence, stress

## Abstract

The inability of an individual to identify, assess, and manage emotions and levels of stress has adverse individual and societal consequences. Previous studies have shown that yoga-based interventions can successfully treat stress, anxiety, and depression, and can enhance emotional control. The aim of the current study was to assess the effect of a specific, intensive, yoga-based intervention, Dynamic Suryanamaskar, on the levels of perceived stress and emotional intelligence in Indian male school students. One hundred and five students with a median age of 17.15 ± 1.42 years were assessed. Practice took place over 12 weeks (*n* = 70 workouts). The Perceived Stress Scale (PSS) questionnaire and the emotional intelligence (EQ) questionnaire, developed for the Indian population, were used to measure stress and emotional levels at the start and end of the study. The Solomon four-group design was used to ensure statistical reliability. The post-study univariate analysis of covariance ANOVA between groups (*p* < 0.001) and the *t*-test for independent samples (*p* < 0.05) indicated that, for those using the Dynamic Suryanamaskar protocol, there was a significant reduction in stress levels and a significant (*p* < 0.01) increase in the levels of emotional intelligence. This study thus provides further evidence of the benefits of the practice of Dynamic Suryanamaskar.

## 1. Introduction

Ancient cultures, such as those found in the Indian subcontinent, have, since antiquity, had programs of physical education that have encompassed areas of health, motor skills, military effectiveness, sports and recreational games, medical therapy, physical rehabilitation, and spiritual development [1]. The system of yoga, itself over five thousand years old [2,3,4,5], forms an important part of such a holistic and integrated approach to human health, stress reduction [6], and physical culture. The techniques and methods of yoga eventually found their way into publications appearing in the fields of physiology and medicine [7,8,9,10]. Additionally, more recently and ever more openly, the introduction of yoga into US schools has been discussed [8,11,12].

### 1.1. Dynamic Suryanamaskar

The intensive yoga exercise routine called Suryanamaskar (SN) has become very popular amongst yoga practitioners of the modern era. Its unique combination of asanas (postures), pranayama (yogic breathing), and dharana (fixing the mind) helps to achieve concentration [3,13,14,15]. The dynamic version of it (DSN) focuses on the intensity with which each round is practiced. Some studies have shown that the maximum pace of DSN performance may impose a physiological burden on the body close to the maximum limit, and that energy expenditure on such occasions may exceed 14.5 MET (VO_2max_ = 90.36%) and may even be up to 16 MET (VO_2max_ = 99.22%) [16].

There are many different versions of Suryanamaskar. Swami Gitananda of Ananda Ashram in southern India mentions 17 [17]; other sources quote 50 styles. These sets of Suryanamaskar sequences were given different names, especially in ancient times, when they formed part of a larger worship and ritual [18], and today they can be found under such titles as ‘Chandra-namaskar’ (Moon Salutations), ‘Guru-namaskar’ (Teacher Salutations), and ‘Hanuman-namaskar’ (Lord Hanuman Salutations), etc. [5,17,19,20]. Although these sets of exercises sometimes differed in their various movements, the Suryanamaskar core remained intact. The sets of exercises constituted a coherent system, which included asanas, pranayama, and fixing of the mind, and the practice was always followed by a short period of relaxation. However, it should be noted that the subjects in the current study did not use DSN as a ritual or worship, but as a hatha yogic practice.

A round of the dynamic form of Suryanamaskar (DSN) includes the set of 12 postures (see Figure A1 in Appendix A) known as the *Rishikesh Series* [13]. The method requires a smooth but rapid transition from one position to the next, and a round should be completed in a time of 7.5 to 8 s. Thus, at the faster of these two rates, in a period of 5 min, a total of 40 rounds would be performed [13]. As far as breathing is concerned, regular DSN, practiced at the first of the four available levels, allows the time for several breaths per posture (usually two to three), while at the second and third levels, the breathing is synchronized with the movements. Only at the fourth, the most advanced level, is there no longer the usual coordination between breathing and the movements; depending on the individual’s capacity, there may be only one to four breaths per round, instead of the six at levels two and three. The sequence of postures in DSN remains exactly the same as that shown in Figure A1 in Appendix A. Some studies have reported that even achieving the advanced age of 80 has not stopped some practitioners from performing 1008 rounds daily [13,19,20]. 

### 1.2. The Physiological and Psychological Effects of Practicing Yoga

An important element in the performance of SN is the controlled breathing, so it is appropriate to compare the effects of DSN with the effects of pranayama (yogic breathing) [21,22]. The study by Crisan [23] demonstrated a reduction in the symptoms of anxiety, a lowering of heart rate, and an increase in galvanic skin resistance in 19 patients with known mental disorders after 8 weeks of the practice of pranayama. The results of these three studies indicate that the effects on the levels of stress as a result of the performance of pranayama or SN over a period of several months are broadly similar in their positive nature [6].

The full practice of DSN also involves another important element from the range of yoga techniques, namely the focusing of attention or fixing of the mind (Sanskrit ‘dharana’). Confirmation of the beneficial effects of this kind of fixing of the mind can be found in the literature relating to the practice of yoga and its impact on the psyche [8,14,24,25,26]. These studies indicate the effectiveness of the various practices of yoga as a means of sustaining physical and mental health and creating a feeling of well-being. The current study has verified those conclusions in the case of DSN specifically.

Properly executed yoga exercises reduce tension not only in the muscles, but also in the mental and emotional spheres, and therefore, are generally considered an effective antidote to stress. Knortz [27] and Ryguła [28] pointed out the relationship between physical (muscle) tension and mental tension. It is this relationship that allows stretching and flexibility exercises to affect the processes of relaxing, toning, rehabilitation, and reconstruction of the body [29]. DSN, owing to its very high physiological intensity (14.5–16 MET), requires a very high level of focused attention. Because the practice engages the practitioner’s whole being, it does not permit the mind to wander or to dwell on negative emotions. The ability to think about what is not happening is a cognitive achievement that comes at an emotional cost. Hence, it is appropriate to compare the effects of DSN practice with the results of using the techniques commonly known as meditation [30].

After using that technique of meditation known as ‘mindfulness’, Kabat-Zinn [31] observed a reduction in the levels of stress, anxiety, depression, and panic. In turn, Ray [32], Woolery [33], and Uebecklacker [34] have demonstrated the beneficial effect of yoga exercises in reducing depression in patients who are diagnosed as having moderate depression or simply as those experiencing a depressed mood. Various studies of the effects of the use of yoga techniques on psychiatric clinic patients diagnosed by the test Profile of Mood States (POMS), have shown similar results in the reduction in depression and in the improvement of mood [35,36,37,38,39]. Other studies have shown that long-term anxiety constitutes a prolonged stressor and can lead to depression [40,41,42].

Although the aim of the present study was not to determine specific levels of depression, it is reasonable to assume that the practice of DSN can prevent such states because of its ability to induce the relaxation effect. Benson [11,12] noted that, in a way that is similar to the psychophysical response to stress, there is an ‘opposite’ relaxation reflex that is activated by simple techniques of mental concentration. Focusing on the mere act of breathing has this effect, no matter the length of inhalation and exhalation, which itself is merely a function of prior training. The use of this simple technique can trigger a relaxation response. Two neurophysiologists from Harvard Medical School, Khalsa and Gould [8], confirmed Benson’s finding that practicing hatha yoga activates the body’s relaxation response. By focusing attention on the breath, or on a thought or a sound, or on the gradual deepening of the breath, a relaxation response occurs automatically, with a consequential reduction in any feeling of stress or anxiety [9]. Sato et al. [43] have found that abdominal breathing (the yogic breathing technique called pranayama uses such a method) has increased the tear meniscus volume in healthy women after only 3 min of practice, thus causing a deactivation of the sympathetic nervous system and thus, a reduction in stress levels. However, it is possible that inexperienced or amateur DSN practitioners do not perform pranayama or dhyana to the extent that they experience the stress-reducing benefits described in the paragraphs above.

The stress level studied in our experiment is a biological term used to describe the consequences of the body’s response to a perceived threat, whether real or imaginary. This reaction includes a sequence of three stages, firstly, the response to the critical situation, during which the secretion of adrenaline occurs, then, a brief period of resistance to the situation, and then exhaustion. Selye defined stress in 1936 as “non-specific response of the body to the threat”. Selye [44] strongly emphasized the distinction between the stressor, or the cause, and the stress that arises as a result/reaction. The perception of danger plays a fundamental role in the arousal of the feeling of stress. Lazarus and Folkman, in 1984, gave a slightly different definition of stress as “the special relationship between the person and the environment, which is assessed as exceeding the person’s capabilities and threatening his/her well-being.” This assessment of the stressful situation contains two elements: the initial assessment that personal safety or well-being is threatened and the secondary assessment of the resources, personal or otherwise, available to deal with the threat. The emphasis on the importance of the assessment of the threat has been supported by the work of Jaskolski and Jaskolska [45]. The latter definition of stress was used for the purpose of this study. 

The emotional intelligence quotient (EQ: emotional quotient), the other metric analyzed in this study, represents a skill, which is defined as the ability to identify, assess, and manage one’s own emotions, those of others, and those of groups. Emotions are one of the main components of human behavior. The ability to control instinctive emotions such as anger or fear and replace them with positive emotions such as sympathy, empathy, compassion, and love, indicates a higher level of emotional health. According to Goleman [46], approximately 80% of success in life is dependent upon emotional competence. Being better able to adapt to different socio-cultural life situations is an important skill that can be measured by a number of personality tests, including those that measure EQ. The results are usually strongly influenced by the cultural context of the individual. Therefore, separate tests were created for different populations and societies [46,47].

There are some studies with randomized controlled trials (RCT) concerning yoga practice and its influence on emotion regulation [48], but there are no studies with RCT that focus on the influence of yoga on EQ. Some non-RCT studies claim that yoga increases EQ [49].

There are dozens of meta-analyses on yoga and its elements’ effects on stress, depression, fatigue, posttraumatic stress disorders, etc. In the context of our study, there are a few meta-analyses worth mentioning. Pascoe et al. [50] studied the relationship between yoga, mindfulness-based stress reduction, and stress-related physiological measures, and they concluded that yoga significantly reduced cortisol, systolic blood pressure, heart rate, and heart rate variability, since yoga asanas appear to be associated with the improved regulation of the sympathetic nervous system and hypothalamic-pituitary-adrenal system in various populations. Another meta-analysis is of a similar significance in the context of our study, that by Breedvelt et al. [51], which studied the effects of meditation, yoga, and mindfulness on depression, anxiety, and stress in university students, and they revealed that the effects were positive but of medium sizes, and when the interventions were compared to active controls, the effects became small. The authors notice that most studies were of poor quality and that the conduct and reporting of studies on meditation, yoga, and mindfulness needs to be more rigorous. In agreement with this statement, we decided to carry out a study using the Solomon four-group design.

### 1.3. Hypotheses

The aim of this study was to determine the effect of the performance of DSN on the level of perceived stress and on the emotional intelligence quotient (EQ) of Indian students. It was hypothesized that DSN training would have a significant positive impact on the ability to cope with stress and a positive effect on the level of emotional intelligence.

Two hypotheses were formulated for testing:Students practicing DSN are better at dealing with stress compared to control subjects.Students practicing DSN increase the level of emotional intelligence in comparison to control subjects.

## 2. Materials and Methods

### 2.1. Experimental Design

The Solomon four-group design was used (Table 1), as it is considered one of the most statistically proper methodologies in experimental research [52,53]. 

The Solomon four-group design combines a standard pre-test and post-test two-group design with a post-test only control design. By using different combinations of tested and untested groups with treatment and control groups, the researcher can ensure that the confounding variables and other extraneous factors have not influenced the results. The Solomon four-group design is a method that addresses some of the challenges of the pre-test and post-test design. It includes two additional control groups that help reduce the influence of confounding variables and allow the researcher to test if there is any effect of the pre-test on the subjects. Therefore, some researchers suggest using ANCOVA in order to treat these measurements as covariates. While the design requires more effort to set up and analyze, this design effectively controls the internal validity issues that may affect the research. It gives the researchers complete control over the variables and allows them to confirm that the pre-test did not influence the results [54,55,56].

In accordance with the Solomon four-group design, four groups of students were chosen on a random basis, each having at least *n* = 25 persons (the exact numbers are shown in Table 2 and Table 3). The test questionnaires were answered twice by Groups I and II, that is, at the beginning of the experiment (Trial-1), and on the last day of the experiment (Trial-2), and answered once by Groups III and IV, that is, on the last day of the experiment. At the start of the experiment, 51 students were subjected to the test, comprising 26 persons from Group I (experimental), who were undertaking the DSN training, and 25 from Group II (control), who were not. Testing at the end of the experiment included a total of 105 students (Groups III and IV had *n* = 27 students each), i.e., all four groups (Table 2). Groups I and III had undertaken the DSN training and Groups II and IV had not.

### 2.2. Respondents

Participants in this study were randomly selected male students with a median age of 17.15 ± 1.42 years (*n* = 105), from the junior and senior years of a high school in the village of Kokamthan in the state of Maharashtra, India. In order to ensure uniformity in the conditions of the experimental groups, the subjects were selected from those students who either were living in a student hostel or whose homes were very close to the school. The diet for all the participants was lacto-vegetarian. The basis of the diet was rice, wheat flour, lentils, vegetables, fruit, milk, and paneer (a non-aged, non-melting soft cheese made by curdling milk with a fruit- or vegetable-derived acid, such as lemon juice). All students ate in the school cafeteria according to the menu prepared and controlled by a school dietician, resulting in a minimal difference in nutrition, as determined by the quality, constituents, and calorie content of the meals The subjects agreed not to ‘disturb’ this norm or to supplement the diet for the period of the study. Before starting the experiment, all subjects signed the official document giving informed consent. All subjects participated on a voluntary basis and with the full acceptance and written consent of their parents. The research was approved by the school director and other school authorities as per the requirements of the Maharashtra State Board of Medicine, and by the Bioethics Research Committee of Jan Dlugosz University in Czestochowa, Poland.

### 2.3. Characteristics of the Study Groups (Mean Values ± SD)

The four groups did not differ significantly in age, weight, height, body fat, and BMI (t-test for independent groups *p* < 0.05), Table 3.

For calculating the total body fat percentage of the subjects, the Sloan–Weir nomogram technique was used. In this technique, two measurements of skin thickness (thigh and subscapular for men) are used. The subscapular skinfold was measured beneath the inferior angle of the left scapula in the direction running obliquely downwards at about a 45° angle from the horizontal. The thigh skinfold was taken on the anterior surface mid-way between the mid-inguinal point and the superior border of the patella, while the knee was flexed at 90°.

### 2.4. Test Procedure

Groups I and III met for practice in the exercise hall between the hours of 6.10 a.m. and 7.00 a.m. Liquids and/or food were not to have been consumed beforehand. The techniques and rules applicable for that week’s level of practice were explained at the start of each week: the basic level for week one, the second level for week two, and the fourth (advanced) level from week three onwards [57,58,59]. Members from Groups I and III took part in all the other activities prescribed by the school curriculum, including physical education classes and recreational sports games, with the exception of the morning's twenty-minute warm-up exercises. Suryanamaskar training was carried out for twelve weeks, six times per week (*n* = 70 workouts; two additional days of holiday were without practice). The physical exercise element of the training lasted a minimum of thirty-five minutes; at the end of each session, five additional minutes were allocated for relaxation in the supine position, the yoga posture known as Shavasana. In the first week of training, subjects were shown the method, given the opportunity to discuss it, and were reminded of the most important rules and principles of performing the DSN technique of twelve asanas; most errors in execution were corrected at that time. In the second week, a thorough review of the DSN technique was made, but only once. In the first month, as an aid to the proper practicing of DSN, the instructor conducted an additional 10 min session of flexibility exercises, primarily to improve the basic technique. While the two experimental groups were undertaking the DSN training, the two control groups participated in the school curriculum morning workout, which consisted of stretching exercises and jogging for about twenty minutes. 

Summing up, the analyzed groups were (following the Solomon four-group design described in the experimental design section above):Group I: DSN morning practice for 12 weeks, 6 times per week; emotional intelligence and stress level measured twice: at the beginning and at the end of the study (repeated measurement);Group II: school curriculum morning workout; emotional intelligence and stress level measured twice: at the beginning and at the end of the study (repeated measurement);Group III: DSN morning practice for 12 weeks, 6 times per week; emotional intelligence and stress level measured once: at the end of the study (one-time measurement);Group IV: school curriculum morning workout; emotional intelligence and stress level measured once: at the end of the study (one-time measurement).

The protocol used is shown in detail in Table A1 in Appendix A.

### 2.5. Research Tools

Verification of the study’s two hypotheses required the assessment of two psychological variables, that is, the level of perceived stress and the level of emotional intelligence quotient (EQ).

The stress level was evaluated by means of the Perceived Stress Scale (PSS) test. It consisted of 10 questions using a five-point Likert scale. Stress is expressed numerically as the sum of the scored points. The test results were expressed as a number ranging from 0 to 40 [60].

The emotional intelligence quotient was evaluated by means of the Chadha test, because the test was developed specifically for Indians [47]. The result given by the test is expressed numerically, as the sum of the scored points obtained from a questionnaire consisting of 15 questions, which are based on socially neutral situations. Five answers are assigned to each question and the result of each answer is expressed as a number from 0 to 20. The overall test score is shown as a number ranging from 0 to 300. The higher the number, the better the EQ [46,61,62].

### 2.6. Statistical Methods

The research results were analyzed using the statistical program SPSS v.28 USA 2021. Basic characteristics of descriptive statistics were calculated, and, to demonstrate the significance of differences between arithmetic variables considered in subsequent studies, the following methods were used for the analysis of variance techniques: univariate analysis of covariance ANCOVA between groups design, univariate analysis of variance ANOVA within group design, and *t*-test for independent samples [55,56,63,64]. The level of statistical significance was set at *p* < 0.05.

According to the statistical guidelines for carrying out analyses in the Solomon four-group design [54], the study first checked whether there was any evidence of pretest sensitization in the groups. The possibility of identifying this is one of the biggest advantages of this experimental design. The test for this is a 2 × 2 between-groups analysis of the variance (ANOVA) on the four post-test scores. The factors are treatment (yes vs. no) and pre-test (yes vs. no). The evidence demonstrating pre-test sensitization is detected by the interaction of the factors. Next, the study performed a two-group analysis of covariance (ANCOVA) on the Trial-2 scores (post-test, second measurement, and at the end of the study) covarying the Trial-1 scores (pre-test, first measurement, and at the beginning of the study), in order to identify the effect of the treatment (DSN practice). 

## 3. Results

### 3.1. Perceived Stress Scale (PSS)

The interaction of the 2 × 2 ANOVA on the four post-test scores turned out to be not significant, with F (1.101) = 0.354, *p* = 0.553, and partial η^2^ = 0.003. We can conclude then that there was no pre-test sensitization effect present, and thus, the effects reported below were most likely to be attributable to the DSN practice.

The univariate analysis of covariance ANCOVA, between-groups design, for Group I and Group II indicated a significant F (1.48) = 19.5, *p* < 0.001, and partial η^2^ = 0.289 reduction in the levels of stress in the experimental Group I. The accompanying covariate was the result measured in Trial-1, and the dependent variable was the result measured in Trial-2 (see Table 4 for the mean and standard deviation figure for each group). 

The difference between the mean result for pre-test/Trial-I scores for Group I and for post-test/Trial-II scores for Group I was statistically significant, with F (1.25) = 6.153, *p* = 0.02, and partial η^2^ = 0.198 (see Table 5).

For Group II, a significant increase was observed in the level of stress F (1.24) = 18.520, *p* < 0.001, partial η^2^ = 0.436 (see Table 6). 

Analysis of *t*-test for independent samples in Group III and Group IV showed a significant difference in the level of stress (Figure 1), t(52) = −2.647, *p* = 0.011, M = −3.5556, SE = 1.34327, lower CI = −6.25102, and higher CI = −0.86009 (see Table 7 for mean results). Group III (practicing DSN) had a lower level of stress in the post-test measurement than Group IV (practicing jogging). 

### 3.2. Emotional Intelligence Quotient (EQ)

The interaction of the 2 × 2 ANOVA on the four post-test scores turned out to be not significant F (1.101) = 0.378, *p* = 0.540, partial η^2^ = 0.004. We can conclude then, that there was no pre-test sensitization effect present, and thus the effects reported below were most likely to be attributable to the DSN practice.

Univariate analysis of covariance ANCOVA, between-groups design, for experimental Group I and control Group II, indicated a significant F (1.48) = 11.133, *p* = 0.002, partial η^2^ = 0.188 increase in the level of emotional intelligence (EQ) in Group I. The accompanying covariate was the result measured in Trial-1, and the dependent variable was the result measured in Trial-2 (see Table 8 for mean results).

The difference between the mean result for pre-test/Trial-I scores for Group I and for post-test/Trial-2 scores for Group I was statistically significant F (1.25) = 5.362, *p* = 0.29, partial η^2^ = 0.177 (see Table 9).

For Group II we observed nonsignificant F (1.48) = 1.720, *p* = 0.202, partial η^2^ = 0.067 differences between the levels of emotional intelligence between pre-test and post-test measurements (see Table 10 for mean results). 

Analysis of t-test for independent samples in Group III and Group IV showed a non-significant difference in the level of stress (Figure 2), t(52) = −2.647, *p* = 0.093, M = 14.2593, SE = 8.33539, Lower CI = −2.46693, Higher CI = 30.98544 (see Table 11 for mean results).

## 4. Discussion

SN is recognized as being one of the various techniques and methods of yoga that is effective in helping practitioners to cope with stress. One of the aims of this study was to determine whether the performance of the dynamic version of Suryanamaskar would have any effect on the stress levels of Indian students. The results confirmed that there had been a positive effect. Despite the significantly greater intensity of the dynamic form of this practice in comparison to static asanas, the stress response was reduced. In Group I (experimental), a significant decrease occurred in the levels of stress, while in Group II, (control) the levels of stress increased. The common factor boosting the levels of tension and stress in all four groups of students was deemed to have been the approaching end of the school-year examinations and matriculation, which were due to take place shortly after the second measurement (Trial-2). It was thus significant that the students in both of the groups practicing DSN (Groups I and III) had a lower stress level of stress than those in the two control groups (Groups II and IV), whose stress levels were increased. However, attributing the stress level rise in the jogging groups to the absence of DSN practice must be regarded as no more than speculative at this stage, and such a conclusion would need to be treated with caution. Further studies are required in order to verify this conjecture. It is also worth drawing attention to the fact that there was a decrease in the stress level variance for Group I subjects in the post-test, a result that is a previously observed effect of yoga practice [65].

Other authors have come to similar conclusions on the effectiveness of yoga exercises in reducing levels of stress, anxiety, and depression [66,67]. In addition, an improvement in subjective feelings of well-being has been demonstrated in another study [68]. Although these latter studies involved yoga, where SN was not the main element of the techniques used, the results are consistent with the results obtained in the current study. 

SN and DSN are consistent with other yogic practices in their application of many of the body positions, sequences of movements, breathing patterns, and concentration techniques found in those other practices. In all of these practices, SN, DSN, and otherwise, the individual exercise is not considered to have been correctly executed unless the relaxing effect has been created. In fact, any practice that may fall under the heading of yoga-based protocol should have the following four components: asanas, pranayamas, attention (or focusing of the mind), and relaxation, Khalsa et al. [9]. Suryanamaskar as a vinyasa also contains these four. Kulmatycki and Burzynski [69] found that in a group of yoga practitioners, the yoga nidra technique (a condition of deep relaxation based on the Bihar School of Yoga, Munger) was effective in decreasing not only anxiety levels, but also levels of anger and depression. Kulmatycki and Burzynski [70] also showed that hatha yoga or postural yoga training reduces both state and trait anxiety, and results in the partial reduction in depressive emotions. Their study also indicated that the more time that was devoted to the practice of yoga, the better was result in dealing with negative emotions. The results of the current study are based on a period of training of 12 weeks, which, as the previous discussion of the results indicated, was sufficient both to reduce stress and help improve the emotional state. Other studies show that lowering levels of anxiety is associated with a decreased resting blood pressure, decreased resting heart rate, and decreased resting respiratory rhythm [8,45,71]. In their 8-week study, McCaffrey [72] showed that the practice of asanas and pranayama contributed to a reduction in the level of stress, and that it was correlated with a reduction in heart rate and systolic and diastolic blood pressure. Other research on pranayama and breath practices have been shown to reduce symptoms of stress, anxiety, insomnia, post-traumatic stress disorder, mass disasters, depression, and attention deficit disorder [73].

Concerning the EQ results that we achieved in this study, we observed a weak increase in the emotional intelligence in the post-treatment measurement in the group of students practicing DSN in comparison with the active group. This goes along with the results presented in the literature [53], as we have previously mentioned; however, this result should be accepted with caution, since the effect was not strong.

Selye [44] describes the body's non-specific reactions to a stimulus or stressor as a stress response occurring independently to the type of stimulus or stressor. In the experiment that constituted the current study, DSN could be considered to have acted as an intensive and long-term stimulus or stressor, which produced a number of non-specific defensive reactions and adaptations in the bodies of the subjects. The results suggest that the DSN training might also contribute to improving the body's overall resistance to stressors. It is possible that DSN improves the adaptation to a non-specific stimulus through the phenomenon known as ‘post-workout supercompensation’ (increasing the possibility of rebuilding ‘excess’ reserves). As a result of this process of rebuilding reserves, the development of the third stage of stress and exhaustion can be significantly delayed [69]. According to some researchers, participation in a 10-week course of yoga improves behavioral control and decreases negative emotions and psychological distress [38,69,74]. Depending on the chosen style or techniques, yoga can be one of the tools capable of triggering positive emotions, creating a good mood or an unusual state of well-being, and of decreasing the level of anxiety or the intensity of a negative stress response [8,9].

To sum up, both hypotheses of the study have been confirmed. Firstly, subjects practicing DSN (Groups I and III) had lowered their perceived stress levels in comparison to the control subjects. Secondly, participants practicing DSN (Group I) had a significant increase in their EQ between the Trial-1 (pre-test) and Trial-2 (post-test), in comparison to those not doing so (the control Group II). Additionally, on a trend level, those practicing DSN from Group III had higher EQ levels than those not doing so from the control Group IV. 

### Limitations

An obvious limitation of the current study is that it has been carried out exclusively on an Indian male population of a young age. We know from medical studies that the general youth population in India has different physiological characteristics in certain respects to Western youth, for instance, a generally lower body mass index (BMI) [75,76]. The practicing of DSN might bring about different outcomes for Western students. On a similar note, we have not studied other cohorts of the Indian population. For example, one of the main reasons why male students only were included in this research was the fact that, especially in rural areas, it would have been unacceptable for foreign male researchers to have administered physical exercises and various tests to female students. Here, there is clearly an opportunity for female researchers.

It would also be advisable (regarding recommendations for future studies) to carry out experiments away from the end-of-term examination period, which might naturally increase stress for participants, and so influence the study results.

Another limitation is that, whereas students from the control group participated in stretching and jogging exercises, there are also other well-known techniques of lowering stress levels, such as mindfulness, slow-breathing techniques, etc. Research shows that more intense aerobic exercising is an important means of lowering stress levels. Both SN and jogging may have a similar energy cost and metabolic intensity [77,78], so there may well be other aspects of DSN that might lower stress levels, such as its intensity, fixing of the mind, respiratory focus, etc. Govindaraj [79] and Stec [13] suggest that emphasis on breath regulation, mindfulness during practice, and importance given to maintenance of postures are some of the elements that differentiate yoga practices from physical exercises. In addition, DSN, owing to its very high physiological intensity (14.5–16 MET), requires a very high level of focused attention [16], and may be one of the most important causes ensuring higher effectiveness then regular low-intensity jogging and stretching. A number of the teenage subjects in this experiment were able to perform, after 12 weeks of training, over 400 rounds continuously, with the best individual reaching about 670 rounds. These other factors may be the reason why it was found in this study that DSN was much more effective in reducing stress levels than simple morning jogging, even when combined with mild stretching. It might be beneficial to design a study involving a couple of control groups, each of which underwent different procedures. It is already known from other research that yoga can be more effective than other techniques [80,81]. However, there is still a shortage of controlled randomized studies in this domain.

## 5. Conclusions

The results obtained in the experimental groups provided a positive answer to the research questions, namely. whether the performance of the dynamic version of Suryanamaskar would have any effect on the stress levels and the emotional intelligence of Indian students. The DSN technique is a form of vigorous physical activity that produces beneficial effects on the mental state. DSN may justifiably be included in the curriculum of yoga techniques, the practice of which results in the raising of the level of emotional intelligence and the lowering of negative emotions, especially the levels of stress and, as a consequence, anger and anxiety. Potential improvement in the ability to adapt to difficult life situations can thus be attributed to the practice of DSN.

## Figures and Tables

**Figure 1 ijerph-20-02845-f001:**
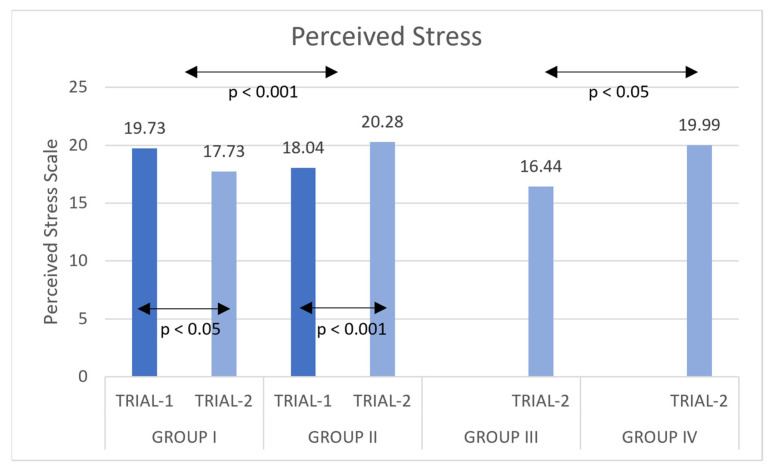
Mean values for the Perceived Stress Scale (x ± SD) pre-test/Trial-1 and post-test/Trial-2 training in DSN.

**Figure 2 ijerph-20-02845-f002:**
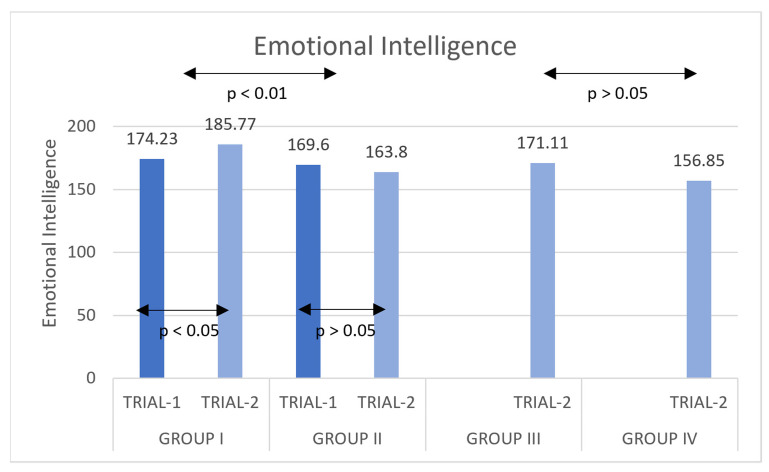
Mean values of the scale emotional intelligence quotient (x ± SD) pre-test/Trial-1 and post-test/Trial-2 training in DSN.

**Table 1 ijerph-20-02845-t001:** Scheme of observation (research).

Group	Selection Method	Trial-1	Treatment	Trial-2
Group I	randomization	O_1_	yes	O_2_
Group II	randomization	O_3_		O_4_
Group III	randomization		yes	O_5_
Group IV	randomization			O_6_

O_1_, O_2_, etc.—observation one, observation one, etc.

**Table 2 ijerph-20-02845-t002:** Number of participants in each group.

Group I	Group II	Group III	Group IV
*n* = 26	*n* = 25	*n* = 27	*n* = 27

**Table 3 ijerph-20-02845-t003:** Demographic characteristics and BMI of the study groups (*n* = 105).

	Age (years)	Weight (kg)	Height (cm)	Fat (%)	BMI (kg/m^2^)
Group I (*n* = 26)	17.05 ± 1.28	50.65 ± 7.45	165.77 ± 6.98	10.89 ± 3.32	18.37 ± 2.07
Group II (*n* =25)	17.35 ± 1.39	54.10 ± 6.36	169.90 ± 7.34	9.60 ± 2.90	18.72 ± 2.14
Group III (*n* = 27)	16.96 ± 1.32	56.48 ± 9.44	169.66 ± 7.60	11.90 ± 5.02	19.55 ± 2.66
Group IV (*n* = 27)	17.26 ± 1.68	53.57 ± 12.82	168.30 ± 6.16	10.70 ± 5.58	18.82 ± 3.88

**Table 4 ijerph-20-02845-t004:** Mean results for Perceived Stress Scale for the ANCOVA model for Groups I and II.

Group	Mean	Std. Deviation	*n*
I	17.731	2.794	26
II	20.280	4.238	25

**Table 5 ijerph-20-02845-t005:** Mean results for Perceived Stress Scale for the ANCOVA model for Group I pre-test/Trial-1 and post-test/Trial-2 measurements.

	Mean	Std. Deviation	*n*
Perceived Stress Scale-Test 1	19.7692	4.41187	26
Perceived Stress Scale-Test 2	17.7308	2.79367	26

**Table 6 ijerph-20-02845-t006:** Mean results for Perceived Stress Scale for the ANCOVA model for Group II pre-test and post-test measurements.

	Mean	Std. Deviation	*n*
Perceived Stress Scale-Test 1	18.0400	4.74763	25
Perceived Stress Scale-Test 2	20.2800	4.23792	25

**Table 7 ijerph-20-02845-t007:** Mean results for Perceived Stress Scale for Groups III and IV post-test measurements.

Group	*n*	Mean	Std. Deviation	Std. Error Mean
III	27	16.4444	4.85429	0.93421
IV	27	20.0000	5.01536	0.96521

**Table 8 ijerph-20-02845-t008:** Mean results for emotional intelligence quotient for the ANCOVA model for Group I and II.

Group	Mean	Std. Deviation	*n*
I	185.7692	30.51859	26
II	163.8000	36.60829	25

**Table 9 ijerph-20-02845-t009:** Mean results for emotional intelligence quotient for the ANCOVA model for Group I pre-test and post-test measurements.

	Mean	Std. Deviation	*n*
Emotional Intelligence Quotient-Test 1	174.2308	42.01648	26
Emotional Intelligence Quotient-Test 2	185.7692	30.51859	26

**Table 10 ijerph-20-02845-t010:** Mean results for emotional intelligence quotient for the ANCOVA model for Group II pre-test and post-test measurements.

	Mean	Std. Deviation	*n*
Emotional Intelligence Quotient-Test 1	169.6000	38.24047	25
Emotional Intelligence Quotient-Test 2	163.8000	36.60829	25

**Table 11 ijerph-20-02845-t011:** Mean results for emotional intelligence quotient for Groups III and IV post-test measurements.

Group	*n*	Mean	Std. Deviation	Std. Error Mean
III	27	171.1111	32.65005	6.28350
IV	27	156.8519	28.45875	5.47689

## Data Availability

The data presented in this study are available on request from the corresponding author. The data are not publicly available due to its sensitivity (psychological data) and the privacy of the study participants.

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
