# Peer review of "Effects of Suryanamaskar, an Intensive Yoga Exercise Routine, on the Stress Levels and Emotional Intelligence of Indian Students"

_ijerph, 2023, doi:10.3390/ijerph20042845_

Round 1

Reviewer 1 Report

These were far too many to list in this way.

A Reviewed version of the article pdf with many Sticky Notes has been sent to the journal to be forwarded to the authors. The Sticky Notes contain a wealth of material that must be attended to in detail. A couple contain positive words of appreciation

Author Response

Reviewer 1

No

Yellow sticker (Reviewer 1 comments)

Authors’ responses

1

The entire manuscript needs the quality of English upgrading. The job needs to be done by a native English speaker with good knowledge of both Science and Yoga. At present it is not well enough written to be published in an international English language journal.

A much earlier version was proof-read by a native speaker, who had knowledge of science and yoga (completed Diploma in Yoga Education 1 year course at Kaivalyadhama). We will use this qualified editor after all the changes are completed.

2

Lord Krishna lived 5 thousand years ago. To use Max Muller's deliberately falsified dating system is unacceptable.

According to the reviewer’s suggestion we have corrected the manuscript and changed two thousand to five thousand.

3

To have fifteen references for a single statement is ridiculous. You only need one or maybe two references that are Authoritative.

According to the reviewer’s suggestion we have corrected the text and  cut down the number of references to three to four.

4

Same comment again. Too many references.

Also, your phrase, 'eventually found' is ambiguous, as the reader does not know if you are referring to Yoga in the Ayurveda literature (which is profound), or in the modern scientific literature, with its dedicated Yoga journals like Yoga Mimamsa.

According to the reviewer’s suggestion we have corrected the text and  cut down the number of references to two to three.

5

In the context of the paragraph as it stands, this last sentence seems irrelevant.

The statement may appear irrelevant to the topic research yet my intention was to show in the first paragraph that yoga has become part of the main stream culture in the West, especially in the United States. According to Prof Sat Bir Khalsa from Harvard Medical School acculturation of a new system like yoga takes place when such basic social systems like the workplace, medical and/or education systems accept it as an integral part.

Hence my inclusion and my suggestion would be to leave it, perhaps after some small modification.

6

This sentence does not make sense when thought about in detail.

I have entered a new statement before this one and deleted the problematic sentences.

7

This phrase is redolent with misunderstanding. The word 'concentration' might mean focussing the attention on the dynamic practice, but it is written as if it refers back to Yoga Dhyana. Dhyana emphatically does NOT involve concentration. That produces frontal gamma in the EEG and fails to produce alpha. Pratyahara and Dhyana result in alpha.

The statement “Owing to its unique combination of asanas (postures), pranayama (yogic breathing) and dhyana (meditation, contemplation) Suryanamaskar helps to achieve concentration

[9,17,25,26]” will be rephrased and it will read: “Owing to its unique combination of asanas (postures), pranayama (yogic breathing)

and focused attention Suryanamaskar helps to achieve concentration

[9,17,25,26]

8

Yoga Anga no. 7, Dhyana, requires prior inward focus of attention, Pratyahara. In Suryanamaskar, the changing positions keep the attention outwards. There is no possibility of Yoga Dhyana occurring.

REMOVE THIS ERROR.

At the beginning of the practice that may certainly be true yet for a seasoned practitioner after a number of years of regular and intense practice THERE IS possibility of dhyana occurring. It happened in my case and in the case of other long-time practitioners. However due to the change of the word dhyana to attention this note has become moot. However, the students – participants of our study – practicing DSN, most probably did not achieve the step of dhyana, because most of the time doing so requires long practice. We have added that in the text.

9

Rituals are rituals. You are referring to a routine of Yoga practice, which is an entirely different matter.

We have modified the text, since the participants of our study did not use DSN as a ritual, but as a practice, as we have now added in the article.

10

This abbreviation is bad writing style.

Better to use the word, 'about'.

It has been changed as suggested.

11

'Can' or 'may', not could

It has been changed accordingly.

12

This has already been stated

This statement has been removed.

13

Again, the relevance of this statement is questionable. Either the Rishikesh version is traditional, and is found in an ancient Yoga text, or it is a modern concoction. Books from the Bihar School contain major errors on every page, and should not be regarded as authoritative. (Believe me, I was horrified to see their quality when I visited Munger.)

It is true that there are no known ancient yoga texts where Suryanamaskar has been mentioned as the set of twelve asanas. Although Suryanamaskar is held by some to be over 5000 years old, it appears to have been adopted into the yoga curriculum by Krishnamacharya [born 1888] at the beginning of the XXth. century in Mysore, so you are right that it may be a modern concoction; furthermore, there are many styles. I have found the name Rishikesh Series applied to this precise sequence of asanas and their names,  as shown in appendix 1, and as used in this study, in the Bihar School of Yoga publication “Surya Namaskara – a Technique of Solar Visualisation”, by Satyananda Swami, Munger, 2006. Satyananda Swami was one of the five chief disciples of Swami Śivananda of Rishikesh, and it is quite possible that some other people may have used that same name for other yoga practices used in the town of Rishikesh (HP).

It is accepted that there certainly are grounds to be concerned about the standards of scholarship of the Bihar School of Yoga and their leader Swami Satyananda, as, indeed, there are about the standards of other widely respected ‘authorities’, such as Krishnamacharya himself. 

However, bearing all these points in mind, and taking into account what has become, whether justifiably or not, an established usage, I should like to suggest that the name Rishikesh Series should continue to be used for this version of Suryanamaskar.

14

But Patanjali Yoga Sutras states very clearly in Pada II that prolonged practice of Pranayama ends in the state of pure consciousness (Pranayama of the 4th kind), and is the way that Swami Rama taught its practice.

There is not the smallest hope of that occurring during DSN, nor afterwards during the period of relaxation, as instructions for pratyahara are absent.

While Patanjali is quite rightly an acknowledged master, it is by no means certain that he would have excluded a priori the possibility of other methods being effective. Those who have practiced Suryanamaskar for prolonged periods, four times per day, and each time for two to three hours, would certainly attest to the effectiveness of such a method.  Those who have not performed the millions of Suryanamaskars which some senior yogis have, might perhaps hesitate before making a definitive judgement.  The experiences described in my book “Dynamic Suryanamaskar-Sun Salutations”, published by SVYASA Yoga University in 2012 and by Motilal Banarsidass in an updated version in 2017, where the two cases of Andre Riehl and Shri Shri Raghavendra Swami are described, might be considered to offer a countervailing argument strong enough to induce such hesitation.

It is possible to contend that, just as Patanjali’s Yoga Sutras are based on experiential knowledge, so are the results claimed for Suryanamaskar practice.

15

You are referring to Dharana, which should never be confused with Dhyana. Maharishi Patanjali very carefully provides definitions of both Pada III, 1&2, and they are completely contrasting.

This kind of persistent misunderstanding of the essence of Yoga makes this paper completely unacceptable for publication, at least in its current form.

I agree that the  word Dharana should be used here and the text will be changed accordingly.

16

This Title Line is totally confusing, and not properly explained. Is the DSN being designated as 'Trial',

or as 'Treatment'. What designates the Assessments?

‘Trial-1’ means the first measurement taken, i.e. the ‘pre-test’, or assessment made before the ‘treatment/DSN intervention’,  and ‘Trial-2’ means the second measurement taken, i.e. the ‘post-test’, or assessment made after the  ‘treatment/DSN intervention’. The use of DSN could be seen as therapeutic in a sense, and that was seen as the justification for using the word ‘treatment’.

17

This sentence is totally confusing unless you know at the beginning what the intended meaning is. Also, it is FAR too long. Split it into at least three parts, making each succinct and easy to understand.

We have completely reconstructed the passage and made it clearer.

18

How can you 'create students'. You are NOT Lord Brahma! The expression is, 'The students were Randomly Divided into 4 Groups.

The statement has been corrected according to the reviewer’s suggestion.

19

But several measures were involved, so the word should be 'measurements'.

The statement has been corrected according to the reviewer’s suggestion.

20

Surely, 26, 26, 26 , & 27 would have been a more equitable distribution.

That was the original and actual distribution, and we can understand that such an arrangement would have been better, as  the reviewer points out, but after a student suddenly had to drop out, unfortunately we had no choice but to accept the new situation as it was.  The dropout of participants is unfortunately a standard situation in longitudinal studies. However, a difference in groups containing 26 and 25 persons is statistically not significant.

21

Paneer is NOT 'cheese'. No more than it is 'cottage cheese' as Indians often try to explain it. .

It is best described as 'freshly precipitated milk solids from Yogurt, known as paneer'

Wikipedia and many other references call it cheese; It is a non-aged, non-melting soft cheese  made by curdling  milk with a fruit- or vegetable-derived acid, such as lemon juice.

To be precise, Indians do not use yogurt but rather milk; for yogurt they would need special bacteria.

The manuscript has been amended to replace farmer’s cheese with “…paneer (a non-aged, non-melting soft cheese made by curdling milk with a fruit- or vegetable-derived acid, such as lemon juice).

22

... this norm, nor to

The statement has been corrected according to the reviewer’s suggestion.

23

Signing the informed consent form could be mentioned here, as well as on line 460.

No study of human subjects is publishable without an IEC Informed Consent, so signing it is an integral aspect of preparing the subjects for participation in the study.

We have restructured the paragraphs. We of course meant here the signing of the informed consent form.

24

There is an Obvious Error in the arithmetic

6 x 12 = 72 (or so I was always told!)

The text has been corrected accordingly. An explanation has been included that two days were given to the students as extra holidays.

It stands now: (n=70 workouts; two additional holidays were without practice).

25

Note the decrease in variance for Group I.

This effect of Yoga practice has been observed previously and is worth commenting on.

The F value, ratio of variances, seems

We are thankful to the reviewer for drawing our attention to this fact. We have added a comment in the “4. Discussion” section about it.

26

This is usually referred to as

'Pre and Post intervention'

We have changed it in the text.

27

But what about Stress Levels for Groups III and IV. It needs comment, whatever the result.

We  describe these stress levels further on in the text, in the next paragraph. We started the discussion section with reference to the hypotheses, and then in next paragraphs we have also described additional results.

28

But this particular concept, 'relaxing effect', was not measured by the assessments made in the study, so you cannot claim it. There are better ways to establish the connection.

The statement has been corrected according to the reviewer’s suggestion.

29

But by what criteria are such a 'relaxing effect' deemed to have occurred. Decreases in Breath Rate and Metabolic Rate must obviously take place immediately after performance of the practice, but that is probably not what you mean.

This statement was intended as a generic statement that any yoga practice should contain four elements, and one of them is relaxation. The other three are asanas, pranayamas and attention.

30

The problem with your use of this concept, hypothesized by Herbert Benson, is that it was shown not to exist in the same way that Selye's stress response exists. It is well understood that stress can become chronic when a person is exposed to too much on a regular basis, Cortisol resistance is well understood. You cannot use this study to claim that it is being overcome.

To avoid the possibility of misrepresentation and confusion arising from these two concepts of the relaxation response as defined by Selye and Benson, the sentence has been deleted.

31

The proximity of these statements imply that you consider results from DSN practice to relate to those from 'Yoga Nidra '. This is obviously untrue.

BESIDES, several techniques are taught under the Name, 'Yoga Nidra', by different Yoga Groups, and you need to specify which particular technique was used in this reference.

The statement has been corrected according to the reviewer’s suggestion.

The Kulmatycki and Burzyński studies were based on the yoga nidra technique as taught by Swami Satyananda from the Bihar School of Yoga, Munger.

32

This paragraph is so badly written that its intended meaning is entirely obscured.

This paragraph has been rewritten according to the reviewer’s suggestions.

33

But there are Masses of such studies in the literature!!! Maybe not yet specifically on those aspects of DSN. BUT that is not what was written. Maybe it is what you meant.

This paragraph has been rewritten according to the reviewer’s suggestions.

34

I thought there were 4 experimental groups

The statement has been corrected according to the reviewer’s suggestion.

Reviewer 2 Report

The manuscript presents an interesting study of the effects of an intensive yoga training regime consisting of 45 min of sun salutations (dynamic yoga sequence) and 5 min relaxation 6 days per week for 12 weeks in young male students. The yoga training was compared to morning stretching exercises and jogging. After 12 weeks, perceived stress decreased and emotional intelligence increased for the yoga participants.

The study is very interesting because it evaluates the effects of one specific yoga exercise/sequence and compares it to an active control that is comparable in intensity and the stretching aspect. However, I have serious concerns regarding the manuscript and data analysis and recommend rejecting the manuscript with the possibility of resubmitting after thorough revision.

My major concern is the lack of any theoretical foundation in the argumentation of the manuscript. Many references seem outdated considering the great number of recent publications on yoga and current theories on yoga are not incorporated at all (e.g., Gard et al. 2014, Front Hum Neurosc; Schmalzl et al. 2018, Consc & Cogn). Furthermore, the different aspects and components of yoga are treated arbitrarily, both in the text and the studies cited. More clarity and precision in the words and concepts used is needed.

One strength of the study is its active control condition – but this is not well argumented or imbedded in the manuscript. It would help to explain the differences between the school curriculum morning workout and DSN- what are commonalities and differences in the procedures and expected effects? Exercise has been proven to have a stress-reducing effect – please elaborate why yoga/DSN should have a better effect (also see Govindaraj et al., 2016, Int Rev Psych).

To my knowledge, the Rishikesh series does not refer to the postures in Suryanamaskar, but is a yoga series on its own. Moreover, I have never heard of the time or round requirements stated in the text, which probably depend on the yoga school it is practiced in. Also, asanas, pranayama and dhyana usually refer to other practices than Suryanamaskar (see Gard et al. 2014 or Schmalzl et al. 2018 for disambiguation). The argumentation and conclusion that DSN is comparable to meditation is very far fetched as this would also apply to any other high intensity training, such as running or cycling.

The argumentation in the introduction is not stringent and the aims/hypotheses of the study do not follow logically from the text. The first section is too broad and arbitrary. There are a lot of meta-analyses regarding the effects of yoga that go well beyond physiology and medicine. Also, how is the introduction of yoga into US schools related to this article? Besides, papers from 1994 and 2000 can hardly be deemly „recent.“ The references cited are too many and refer mostly to books, please reduce them to most important ones.

In addition, there have been over 100 meta-analyses on yoga and its diverse effects, including depression, stress and anxiety. Please refer to these when reflecting yoga’s effect on depression, stress and anxiety. Likewise, why is there so much text on depression and anxiety, if these conditions are irrelevant to/not measured in the study? Neurosis is an outdated term ad there have been numerous studies investigating the effects of pranayama since 1988 (see Brown, Gerbarg & Muench, 2013). The conclusions and features of the current study should not be mentioned in the introduction. The descriptions of stress and emotional intelligence are very basic and unrelated to yoga or DSN. Please explain in the text why these skills should be improved by these kinds of interventions. Your hypotheses should be the logical consequence of your argumentation in the introduction.

Explain why you opted for the Solomon Four-Group Design- did you expect significant amounts of pretest sensitization? Regarding the data analysis, if there was no sensitization effect, it would be advisable to conduct a 2x2 between-group repeated measurements ANOVA instead of an ANCOVA.

Please present the results in the same order as your hypotheses. The data in the tables is redundant to the data presented in Figures 1 and 2. Preferably, report only the figures. Use standard abbreviations for statistical parameters (MD for mean difference, CI for confidence interval). In one section, level of stress is written instead of emotional intelligence.

In the discussio, you cannot state that groups 3 and 4 increased or decreased their levels of stress as there was no pretest. When discussing your results, please refer to the magnitude of yoga research that is available, not only a few arbitrary studies. Likewise, yoga consists of multiple components and it is unclear how these components exert their effects (see Matko et al. 2021, OBM Int Compl Med). Thus, mixing different findings of interventions with differing yoga components does not seem plausible or conductive to this study and its main strength.

There are some minor language issues.

Overall, this manuscript would benefit from a greater clarity regarding key terms of yoga, a more stringent argumentation and a more up-to-date reference list. Then it could provide a valuable contribution to current yoga research.

Author Response

Review Report Form

Open Review

English language and style

( ) English very difficult to understand/incomprehensible
( ) Extensive editing of English language and style required
(x) Moderate English changes required
( ) English language and style are fine/minor spell check required
( ) I don't feel qualified to judge about the English language and style

Yes

Can be improved

Must be improved

Not applicable

Does the introduction provide sufficient background and include all relevant references?

( )

( )

(x)

( )

Are all the cited references relevant to the research?

( )

( )

(x)

( )

Is the research design appropriate?

( )

(x)

( )

( )

Are the methods adequately described?

( )

(x)

( )

( )

Are the results clearly presented?

( )

(x)

( )

( )

Are the conclusions supported by the results?

( )

( )

(x)

( )

Comments and Suggestions for Authors

The manuscript presents an interesting study of the effects of an intensive yoga training regime consisting of 45 min of sun salutations (dynamic yoga sequence) and 5 min relaxation 6 days per week for 12 weeks in young male students. The yoga training was compared to morning stretching exercises and jogging. After 12 weeks, perceived stress decreased and emotional intelligence increased for the yoga participants.

The study is very interesting because it evaluates the effects of one specific yoga exercise/sequence and compares it to an active control that is comparable in intensity and the stretching aspect. However, I have serious concerns regarding the manuscript and data analysis and recommend rejecting the manuscript with the possibility of resubmitting after thorough revision.

My major concern is the lack of any theoretical foundation in the argumentation of the manuscript. Many references seem outdated considering the great number of recent publications on yoga and current theories on yoga are not incorporated at all (e.g., Gard et al. 2014, Front Hum Neurosc; Schmalzl et al. 2018, Consc & Cogn). Furthermore, the different aspects and components of yoga are treated arbitrarily, both in the text and the studies cited. More clarity and precision in the words and concepts used is needed.

One strength of the study is its active control condition – but this is not well argumented or imbedded in the manuscript. It would help to explain the differences between the school curriculum morning workout and DSN- what are commonalities and differences in the procedures and expected effects? Exercise has been proven to have a stress-reducing effect – please elaborate why yoga/DSN should have a better effect (also see Govindaraj et al., 2016, Int Rev Psych).

To my knowledge, the Rishikesh series does not refer to the postures in Suryanamaskar, but is a yoga series on its own. Moreover, I have never heard of the time or round requirements stated in the text, which probably depend on the yoga school it is practiced in. Also, asanas, pranayama and dhyana usually refer to other practices than Suryanamaskar (see Gard et al. 2014 or Schmalzl et al. 2018 for disambiguation). The argumentation and conclusion that DSN is comparable to meditation is very far fetched as this would also apply to any other high intensity training, such as running or cycling.

I have found this name Rishikesh Series for the sequence of asanas and their names used in the Suryanamaskar version adopted for this study in the publication prepared by the well-known Bihar School of Yoga and authored by Satyananda Swami “Surya Namaskara – a Technique of Solar Visualisation” Munger: Yoga Publication Trust: Munger, 2006. Satyananda Swami was one of the 5 chief disciples of Swami Śivananda of Rishikesh and it is quite possible that some other people may have used that name for different yoga practices. I have done quite extensive research on the Suryanamaskar tradition in various states of India and my findings have been published in the book which appeared with other publications by SVYASA Yoga University (Bangalore) under the title “Dynamic Suryanamaskar –Sun Salutations” in 2012 and later on in the  new updated edition under the title “Suryanamaskar-Sun Salutations”, published by Motilal Banarsidass in 2017.

I have looked at the articles by Gard et al. 2014 or Schmalzl et al. 2018 and they refer to the general impact of yogic practices and not specifically to Suryanamaskar, which in many ways is quite different from the main stream of postural yoga. It is called by Krishnamacharya and his close followers, like Patabhi Joise, BKS Iyengar and Desikachar, a vinyasa, by which they understand a more dynamic combination of bodily postures with breathing patterns. In fact it rather could be called a combination of vyayamas (not asanas) with a certain breathing pattern (ujjayi). In my books I have outlined four levels of Suryanamaskar practice, each one distinctly different, in addition to three ways of ayurvedic practising in order to balance the three doshas: vata, pitta and kapha. In this study we have used the 4th level of practice, which is very rigorous and intense, hence my suggestion of naming it ‘Dynamic Suryanamaskar’ [lines 55-72 of my manuscript]. This 4th level of practice of Suryanamaskar practice was inspired by a very important text for perhaps more proper  interpretation of intensity and rigorousness of hatha yogic practices. It comes from the discovered but unpublished text “Asanayoga – Hathabhasya-paddhati” authored by Kapalakurantaka [Kapalakurantaka is one of the seventeen siddhas mentioned in the manuscript Hathapradipika sometimes erroneously called Hatha-yoga-pradipika] and reported by Devnath P during the 1st International Conference on the Revival  of Traditional Yoga in Lonavla in January 2006 – this text says that in order for the hatha practices to be effective they may have to be repeated thousands of times with great intensity ….

Also one needs to remember that Suryanamaskar appears to have been adopted intothe  yoga curriculum only about 100-150 years ago though it has been in existence from Vedic times.

And in the appendices of the edition published by Motilal Banarsidass my friend and scholar, the late Wlodek Łagodzki described in detail a very advanced version called Purna-Suryanamaskar; an intensive practice using the Mahayoga method.

Suryanamaskar consist of several aspects. There are certain postures and breathing patterns which must be performed with an attentive mind, and should be followed by a period of bodily relaxation. According to a well-known researcher, Prof Sat Bir Khalsa from Harvard Medical School (USA), any yogic practice must contain four elements: asanas, pranayama, a focused mind (what he calls meditation, without going into details of Samyama), and relaxation. The tradition of Mahayoga in its Purna-Suryanamaskar form adds to this requirement proper diet [sattvic & mitahara]. That attentive mind can be called in yogic terms, dharana; I have used the term ‘meditation’, which is more widely understood by non-experts, and I did not mean, technically, dhyana, at least not in the initial stages. However, the subjective chapters in my book refer to the practical experiences of the long-term Suryanamaskar practitioners, which would suggest that the state of dhyana (and higher) can be attained through Suryanamaskar.

I have no objective data but subjective reports from some advanced yoga masters and yoga practitioners, reports which explain why Suryanamaskar may be called dynamic meditations. And this should not be a surprise, because a number of outstanding, world-class sportsmen, performers, musician, and artists also report entering the  so-called  Flow State, and that usually comes when physical effort reaches some maximum level.

Hence in India there are sects of religious practitioners, which practice only Suryanamaskar and claim that it is the complete method [sort of psychosomatic cross-fitness] to reach God.

The argumentation in the introduction is not stringent and the aims/hypotheses of the study do not follow logically from the text. The first section is too broad and arbitrary. There are a lot of meta-analyses regarding the effects of yoga that go well beyond physiology and medicine. Also, how is the introduction of yoga into US schools related to this article? Besides, papers from 1994 and 2000 can hardly be deemly „recent.“ The references cited are too many and refer mostly to books, please reduce them to most important ones.

Comparatively there are very few research papers related directly to the Suryanamaskar technique and even fewer to the very little known intense and vigorous version for which I have coined the name Dynamic Suryanamaskar because of its intense and rigorous practice (level 4) [Stec K. Dynamic Suryanamaskar – Sun Salutations. India: Swami Vivekananda Yoga Prakashan, Bangalore 2012, 296.]. Hence here comes forth the uniqueness of this investigation and my difficulty in locating more recent bibliography that would directly relate to the Dynamic Suryanamaskar  practice and not to yoga in general.

Relation of introducing yoga to US schools: The statement may appear irrelevant to the topic researched yet my intention was to show in the first paragraph that yoga has become a part of the main stream culture in the West, especially in the United States. According to Prof Sat Bir Khalsa from Harvard Medical School acculturation of a new system like yoga takes place when such basic social systems like the workplace, medical and/or education systems accept it as an integral part. Hence my inclusion and my suggestion would be to leave it, perhaps after some small modification.

In addition, there have been over 100 meta-analyses on yoga and its diverse effects, including depression, stress and anxiety. Please refer to these when reflecting yoga’s effect on depression, stress and anxiety. Likewise, why is there so much text on depression and anxiety, if these conditions are irrelevant to/not measured in the study? Neurosis is an outdated term ad there have been numerous studies investigating the effects of pranayama since 1988 (see Brown, Gerbarg & Muench, 2013). The conclusions and features of the current study should not be mentioned in the introduction. The descriptions of stress and emotional intelligence are very basic and unrelated to yoga or DSN. Please explain in the text why these skills should be improved by these kinds of interventions. Your hypotheses should be the logical consequence of your argumentation in the introduction.

Added suggested citation

Richard P. Brown MD, Patricia L. Gerbarg MD, Muench F.  Breathing practices for treatment of psychiatric and stress-related medical conditions. Psychiatr Clin North Am. 2013 Mar;36(1):121-40. doi: 10.1016/j.psc.2013.01.001

Explain why you opted for the Solomon Four-Group Design- did you expect significant amounts of pretest sensitization? Regarding the data analysis, if there was no sensitization effect, it would be advisable to conduct a 2x2 between-group repeated measurements ANOVA instead of an ANCOVA.

We had expected a sensitization effect of the kind that is encountered in cases that include repeated processes of measurement. This phenomenon is found especially with questionnaires, where subjects can remember their earlier responses. In addition, this Solomon experimental design is described as the best and most accurate for social studies regardless of synthetizing effects. Most other researchers do not use it because of the doubling-up of the costs. We wanted to have a methodologically solid design. We have strongly rephrased the subsection on method, where we describe the Solomon Design.

Please present the results in the same order as your hypotheses. The data in the tables is redundant to the data presented in Figures 1 and 2. Preferably, report only the figures. Use standard abbreviations for statistical parameters (MD for mean difference, CI for confidence interval). In one section, level of stress is written instead of emotional intelligence.

We have corrected and presented the results in the order of our hypothesis. The data in the tables present means and standard deviations, which need to be presented according to APA reporting standards. We are aware of the fact that these results could be presented in the body of the text, however, we believe that presenting them in a table makes them more easily accessible for the reader. We have changed the abbreviations for statistical parameters. We have also corrected the mistake of referring to stress instead of EQ in two tables.

In the discussion, you cannot state that groups 3 and 4 increased or decreased their levels of stress as there was no pretest. When discussing your results, please refer to the magnitude of yoga research that is available, not only a few arbitrary studies. Likewise, yoga consists of multiple components and it is unclear how these components exert their effects (see Matko et al. 2021, OBM Int Compl Med). Thus, mixing different findings of interventions with differing yoga components does not seem plausible or conductive to this study and its main strength.

The overlooked error that groups 3 and 4 increased or decreased their levels of stress has been now corrected and the statement modified accordingly. In addition, we have added a statement based on the  book of Khalsa et al. [21] as follows and inserted that in line:

“In fact, any practice which may be called YBP (Yoga Based Protocol) should have the following four components: asanas, pranayamas, attention (or focusing of the mind), and relaxation (Khalsa et al. [21]). Suryanamaskar as a vinyasa also contains these four components”.

There are some minor language issues.

The manuscript has again been corrected by a native speaker.

Overall, this manuscript would benefit from a greater clarity regarding key terms of yoga, a more stringent argumentation and a more up-to-date reference list. Then it could provide a valuable contribution to current yoga research.

Submission Date

30 November 2022

Date of this review

20 Dec 2022 14:30:26

Reviewer 3 Report

The introduction is generally well-researched and well-structured. Your research is certainly relevant at present, especially with the rise in mental health in school children. However, in reading the portion of the results, I have some concerns. I hope those suggestions help to further improvements in the manuscript:

Line 52 - In  sum, (do you mean summary?) Please can you write the full word –

In sum, they created a coherent system, which includes asanas, pranayama and focusing of the mind, which is always followed by a short period of relaxation.  

Lines 139 and 140 consider changing the word “this” to “The”

The emphasis on the importance of the assessment has been supported by the work of Jaskolski and Jaskolska [51]. The latter definition of stress was used for the purpose of this study.

Line 152: The statement below is slightly ambiguous – do you mean in your study or the other studies you are referring to?

Therefore separate tests were created for different populations and societies.

Line 159 – consider changing the word people to Students as it is students you are hypothesizing and not the general public

Line 161: do you mean DSN students rather than DSN practitioners?

Throughout the script, authors refer to the participants as “students” or “subjects” or “practitioners” there is no consistency. Please can you address the participants as Students rather than subjects?

The authors mention that participants were all male – can you give the rationale for this, please? Why were they all male? Was this because it was an all-male school?

Line 356 – 359 authors mention Practitioner? And then students again, please can you clarify whom you mean? Please stick to addressing the participants as students to avoid confusion.

Suryanamaskar is recognized as being one of the various techniques and methods of yoga that are effective in helping practitioners to cope with stress. One of the aims of this study was to determine whether the performance of the dynamic version of Suryanamaskar would have any effect on the stress levels of Indian students.

Limitation – I think the authors can add that the study was only conducted on male students.

Also, regarding future recommendations – would it be appropriate to say that future experiments should be done away from the normal end-of-term exams – as it clearly causes heightened stress whereby changing the study results? 

Author Response

Review Report Form 

Open Review 

( ) I would not like to sign my review report 
(x) I would like to sign my review report 

English language and style 

( ) English very difficult to understand/incomprehensible 
( ) Extensive editing of English language and style required 
(x) Moderate English changes required 
( ) English language and style are fine/minor spell check required 
( ) I don't feel qualified to judge about the English language and style 

Yes 

Can be improved 

Must be improved 

Not applicable 

Does the introduction provide sufficient background and include all relevant references? 

(x) 

( ) 

( ) 

( ) 

Are all the cited references relevant to the research? 

(x) 

( ) 

( ) 

( ) 

Is the research design appropriate? 

(x) 

( ) 

( ) 

( ) 

Are the methods adequately described? 

(x) 

( ) 

( ) 

( ) 

Are the results clearly presented? 

( ) 

(x) 

( ) 

( ) 

Are the conclusions supported by the results? 

(x) 

( ) 

( ) 

( ) 

Comments and Suggestions for Authors 

The introduction is generally well-researched and well-structured. Your research is certainly relevant at present, especially with the rise in mental health in school children. However, in reading the portion of the results, I have some concerns. I hope those suggestions help to further improvements in the manuscript: 

Line 52 - In  sum, (do you mean summary?) Please can you write the full word – 

In sum, they created a coherent system, which includes asanas, pranayama and focusing of the mind, which is always followed by a short period of relaxation.   

We have changed the passage, according to the Reviewer’s comment.  

  

Lines 139 and 140 consider changing the word “this” to “The” 

The emphasis on the importance of the assessment has been supported by the work of Jaskolski and Jaskolska [51]. The latter definition of stress was used for the purpose of this study. 

We have changed the passage, according to the Reviewer’s comment. 

  

Line 152: The statement below is slightly ambiguous – do you mean in your study or the other studies you are referring to? 

Therefore separate tests were created for different populations and societies. 

We have moved the two references from the previous sentence in order to indicate that the statement refers to the other studies. We are grateful to the Reviewer for turning our attention to this passage. 

Line 159 – consider changing the word people to Students as it is students you are hypothesizing and not the general public 

We have changed the passage, according to the Reviewer’s comment. 

Line 161: do you mean DSN students rather than DSN practitioners? 

We have changed the passage, according to the Reviewer’s comment, to “students practicing DSN”. 

Throughout the script, authors refer to the participants as “students” or “subjects” or “practitioners” there is no consistency. Please can you address the participants as Students rather than subjects? 

We have changed the reference to students in the whole paper, according to the Reviewer’s comment. 

The authors mention that participants were all male – can you give the rationale for this, please? Why were they all male? Was this because it was an all-male school? 

 We have now provided in the Limitations section an explanation as to why only male students were used. One of the main reasons why female students were not included was the fact that, especially in the rural areas of India, it would be unacceptable for foreign male researchers to administer physical exercises and various tests to such a female group. Another reason was that the school management gave us permission to conduct this research only in the high school where boys were admitted.   

Line 356 – 359 authors mention Practitioner? And then students again, please can you clarify whom you mean? Please stick to addressing the participants as students to avoid confusion. 

“Suryanamaskar is recognized as being one of the various techniques and methods of yoga that are effective in helping practitioners to cope with stress. One of the aims of this study was to determine whether the performance of the dynamic version of Suryanamaskar would have any effect on the stress levels of Indian students.” 

We have changed the passage, according to the Reviewer’s comment. 

Limitation – I think the authors can add that the study was only conducted on male students. 

 We have now added that to the Limitations section. 

Also, regarding future recommendations – would it be appropriate to say that future experiments should be done away from the normal end-of-term exams – as it clearly causes heightened stress whereby changing the study results?  

We have now added that to the recommendations. 

  

Submission Date 

30 November 2022 

Date of this review 

14 Dec 2022 19:29:53 

Round 2

Reviewer 1 Report

Revisions are well done, thank you.

MS should be published

Author Response

REVIEWER 1 (round 2)

Review Report Form

Open Review

English language and style

( ) English very difficult to understand/incomprehensible
( ) Extensive editing of English language and style required
( ) Moderate English changes required
( ) English language and style are fine/minor spell check required
(x) I don't feel qualified to judge about the English language and style

Yes

Can be improved

Must be improved

Not applicable

Does the introduction provide sufficient background and include all relevant references?

(x)

( )

( )

( )

Are all the cited references relevant to the research?

(x)

( )

( )

( )

Is the research design appropriate?

(x)

( )

( )

( )

Are the methods adequately described?

(x)

( )

( )

( )

Are the results clearly presented?

(x)

( )

( )

( )

Are the conclusions supported by the results?

(x)

( )

( )

( )

Comments and Suggestions for Authors

Revisions are well done, thank you.

We are delighted that the changes we have implemented are acceptable to the reviewer.

MS should be published

Submission Date

30 November 2022

Date of this review

13 Jan 2023 20:08:58    

Reviewer 2 Report

Thank you for the revised manuscript. The authors have addressed some, but unfortunately not all of my suggestions. The methods, results and limitation sections have improved as well as the English language.

Although the introduction and discussion section were improved, they are still my major concern. I agree that there is a scarcity of studies on specific components or exercises of yoga. However, this does not mean that you can simply equate Suryanamaskara to pranayama (p. 3, l. 112-13) or meditation (p. 3, l. 133-34). This is very inappropriate. SN surely shares some aspects of these practices, but this does not imply they are the same.

Some of my earlier suggestions concerning the introduction were not addressed at all by the authors:

-       One strength of the study is its active control condition – but this is not well argumented or imbedded in the manuscript. It would help to explain the differences between the school curriculum morning workout and DSN- what are commonalities and differences in the procedures and expected effects? Exercise has been proven to have a stress-reducing effect – please elaborate why yoga/DSN should have a better effect (also see Govindaraj et al., 2016, Int Rev Psych).

-       There have been over 100 meta-analyses on yoga and its diverse effects, including depression, stress and anxiety. Please refer to these when reflecting yoga’s effect on depression, stress and anxiety.

-       Likewise, why is there so much text on depression and anxiety, if these conditions are irrelevant to/not measured in the study?

-       Neurosis is an outdated term.

-       The conclusions and features of the current study should not be mentioned in the introduction.

-       The descriptions of stress and emotional intelligence are very basic and unrelated to yoga or DSN. Please explain in the text why these skills should be improved by these kinds of interventions.

-       Your hypotheses should be the logical consequence of your argumentation in the introduction.

Presenting the results in tables is reasonable, however, presenting the same numbers in the tables and figures is redundant. In this case, you should remove the mean per group from the figures. In addition, you should present all numbers from tables 4 to 6 (and 8 to 10) in one table as they are redundant.

In the discussion, you still state that groups 3 and 4 increased/decreased their levels of stress (p. 11, l. 438-440). Instead, state that group 3 had lower levels of stress than group 4 as you cannot make any conclusions on increase or decrease based on your data.

In the next paragraph you compare your study to a study by Surdarshan, although the two studies examined two related, but nevertheless distinct constructs, i.e., anxiety and stress. You cannot “lump together” different psychological constructs and claim they are basically the same thing.

In addition, you do not discuss your results on emotional intelligence and draw comparisons to earlier studies.

Please be more clear and accurate on the terms you use, both concerning psychological variables and yoga components. At the moment, it seems a bit arbitrary. This also refers to the references you cite.

Author Response

REVIEWER 2 (round 2)  

Review Report Form

Open Review

English language and style

( ) English very difficult to understand/incomprehensible
( ) Extensive editing of English language and style required
( ) Moderate English changes required
(x) English language and style are fine/minor spell check required
( ) I don't feel qualified to judge about the English language and style

Yes

Can be improved

Must be improved

Not applicable

Does the introduction provide sufficient background and include all relevant references?

( )

( )

(x)

( )

Are all the cited references relevant to the research?

( )

( )

(x)

( )

Is the research design appropriate?

(x)

( )

( )

( )

Are the methods adequately described?

(x)

( )

( )

( )

Are the results clearly presented?

( )

(x)

( )

( )

Are the conclusions supported by the results?

( )

( )

(x)

( )

Comments and Suggestions for Authors

Thank you for the revised manuscript. The authors have addressed some, but unfortunately not all of my suggestions. The methods, results and limitation sections have improved as well as the English language.

We appreciate the fact that the Reviewer has noticed various improvements to the main text. We hope that the missing changes implemented with the round 2 will be acceptable. We are also happy that the English language updates suggested by the native speaker are acceptable.

Although the introduction and discussion section were improved, they are still my major concern. I agree that there is a scarcity of studies on specific components or exercises of yoga. However, this does not mean that you can simply equate Suryanamaskara to pranayama (p. 3, l. 112-13) or meditation (p. 3, l. 133-34). This is very inappropriate. SN surely shares some aspects of these practices, but this does not imply they are the same.

We have indicated that physiologically  pranayama and SN results are broadly speaking similar, but we do not claim they are identical. In fact not only we have quoted the similar views expressed by some yoga experts but most importantly we have based that statement on the official publication of the Central Government Ministry of AYUSH which states: “Surya Namaskara is a complete Sadhana, spiritual practice, in itself for it includes Asana, Pranayama, Mantra and meditation techniques”. Also Krishnamacharya branch of yoga tradition, which includes prominent teachers such as Deshikacharia, BKS Iyengar and Patabhi Joise, clearly describe pranayama UJJAYI as a vital part of their SN technique.

Nevertheless, we agree with the Reviewer that DSN may not always include the pranayama part (especially if performed in an inappropriate or amateurish way), therefore we have added the following passage in the text on page 3: “However, it is possible that inexperienced or amateur DSN practitioners do not perform pranayama or dhyana to the extent that they experience the stress-reducing benefits described in the paragraphs above.”

Some of my earlier suggestions concerning the introduction were not addressed at all by the authors:

-       One strength of the study is its active control condition – but this is not well argumented or imbedded in the manuscript. It would help to explain the differences between the school curriculum morning workout and DSN- what are commonalities and differences in the procedures and expected effects? Exercise has been proven to have a stress-reducing effect – please elaborate why yoga/DSN should have a better effect (also see Govindaraj et al., 2016, Int Rev Psych).

Analysis presented in the article applies to this study, the following text has been incorporated “Emphasis on breath regulation, mindfulness during practice, and importance given to maintenance of postures are some of the elements which differentiate yoga practices from physical exercises.” Also another reference described in the table 1 in the book “Suryanamaskar-Sun Salutations” and titled “Characteristics of conventional exercise  compared to Yoga and Suryanamaskar for an  accomplished practitioner in each case” was added and is discussed.

A statement was added in the main text: Govindaraj [76] and Stec [13] suggest that emphasis on breath regulation, mindfulness during practice, and importance given to maintenance of postures are some of the elements, which differentiate yoga practices from physical exercises.

-       There have been over 100 meta-analyses on yoga and its diverse effects, including depression, stress and anxiety. Please refer to these when reflecting yoga’s effect on depression, stress and anxiety.

        We are now referring to some meta-analyses in the text, where appropriate. We have added the following passage to the Introduction: “There are dozens of meta-analyses on yoga and its elements’ effects on stress, depression, fatigue, posttraumatic stress disorders etc. In the context of our study, there are a few meta-analyses worth mentioning. Pascoe et al. [50] studied the relationship between yoga, mindfulness-based stress reduction and stress-related physiological measures, and they concluded that yoga significantly reduced cortisol, systolic blood pressure, heart rate, heart rate variability, since yoga asanas appear to be associated with improved regulation of the sympathetic nervous system and hypothalamic-pituitary-adrenal system in various populations. Another meta-analysis is of a similar significance in the context of our study – Breedvelt et al. [51] studied the effects of meditation, yoga, and mindfulness on depression, anxiety, and stress in tertiary education students, and they revealed that the effects were positive but of medium sizes, and when the interventions were compared to active controls the effects became small. The authors notice that most studies were of poor quality and that the conduct and reporting of studies on meditation, yoga, and mindfulness needs to be more rigorous. In agreement with this statement, we decided to carry out a study using the Solomon Four-Group Design.”

-       Likewise, why is there so much text on depression and anxiety, if these conditions are irrelevant to/not measured in the study?

      We wanted to show different psychological aspects that yoga practice can influence.

-       Neurosis is an outdated term.

According to the reviewer’s suggestion we have corrected the manuscript and changed the word ‘neurosis’  to ‘mental disorders’..

-       The conclusions and features of the current study should not be mentioned in the introduction.

The text in the Introduction “A number of the teenage subjects in this experiment were able to perform, after 12 weeks of training, over 400 rounds continuously, with the best individual reaching about 670 rounds.” has been shifted to the discussion section.

-       The descriptions of stress and emotional intelligence are very basic and unrelated to yoga or DSN. Please explain in the text why these skills should be improved by these kinds of interventions.

      We have added a description of studies analysing the connection of yoga asanas with stress reduction and EI potential enhancement.

The relation between yoga and EI is now described in the following way: “There are some studies with Randomized Controlled Trials (RCT) concerning yoga practice and its influence on emotion regulation [48] but no studies with RCT and the influence of yoga on EQ. Some non-RCT studies claim that yoga increases EQ [49].”

-       Your hypotheses should be the logical consequence of your argumentation in the introduction.

We hope that now the introduction with its clearly stated hypothesis clearly displays the logic.

Presenting the results in tables is reasonable, however, presenting the same numbers in the tables and figures is redundant. In this case, you should remove the mean per group from the figures. In addition, you should present all numbers from tables 4 to 6 (and 8 to 10) in one table as they are redundant.

We were following another Reviewer’s suggestions of adding tables. We believe that the presence of group means in the figures makes it more readable for the article audience. Tables 4-10 present results for different groups and measurements, and we describe the results in the text, following the presentation of the tables, therefore we believe it is a better rendition of results for the readers.

In the discussion, you still state that groups 3 and 4 increased/decreased their levels of stress (p. 11, l. 438-440). Instead, state that group 3 had lower levels of stress than group 4 as you cannot make any conclusions on increase or decrease based on your data.

According to the reviewer’s suggestion we have corrected the manuscript and changed the word ‘increased/decreased  to ‘lower/higher levels..

In the next paragraph you compare your study to a study by Surdarshan, although the two studies examined two related, but nevertheless distinct constructs, i.e., anxiety and stress. You cannot “lump together” different psychological constructs and claim they are basically the same thing.

We have deleted this paragraph containing Sudarshan experiment  in the text.

In addition, you do not discuss your results on emotional intelligence and draw comparisons to earlier studies.

We have added the following passage to the Discussion: “Concerning the EQ results that we achieved in this study, we observed a weak increase in the emotional intelligence in the post-treatment measurement in the group of students practicing DSN in comparison with the active group. This goes along the results presented in the literature [49], as we have previously mentioned, however, this result should be accepted with caution, since the effect was not strong.”

Please be more clear and accurate on the terms you use, both concerning psychological variables and yoga components. At the moment, it seems a bit arbitrary. This also refers to the references you cite.

submission Date

30 November 2022

Date of this review

10 Jan 2023 13:26:37
